# Polarized subcellular activation of Rho proteins by specific ROPGEFs drives pollen germination in *Arabidopsis thaliana*

Alida Melissa Bouatta, Franziska Anzenberger, Lisa Riederauer, Andrea Lepper, Philipp Denninger ⓘ *

Technical University of Munich, School of Life Sciences, Plant Systems Biology, Freising, Germany

* philipp.denninger@tum.de

## Abstract

During plant fertilization, excess male gametes compete for a limited number of female gametes. The dormant male gametophyte, encapsulated in the pollen grain, consists of two sperm cells enclosed in a vegetative cell. After reaching the stigma of a compatible flower, quick and efficient germination of the vegetative cell to a tip-growing pollen tube is crucial to ensure fertilization success. Rho of Plants (ROP) signaling and their activating ROP Guanine Nucleotide Exchange Factors (ROPGEFs) are essential for initiating polar growth processes in multiple cell types. However, which ROPGEFs activate pollen germination is unknown. We investigated the role of ROPGEFs in initiating pollen germination and the required cell polarity establishment. Of the five pollen-expressed ROPGEFs, we found that GEF8, GEF9, and GEF12 are required for pollen germination and male fertilization success, as *gef8;gef9;gef12* triple mutants showed almost complete loss of pollen germination *in vitro* and had a reduced allele transmission rate. Live-cell imaging and spatiotemporal analysis of subcellular protein distribution showed that GEF8, GEF9, and GEF11, but not GEF12, displayed transient polar protein accumulations at the future site of pollen germination minutes before pollen germination, demonstrating specific roles for GEF8 and GEF9 during the initiation of pollen germination. Furthermore, this novel GEF accumulation appears in a biphasic temporal manner and can shift its location laterally. We showed that the C-terminal domain of GEF8 and GEF9 confers their protein accumulation and demonstrated that GEFs locally activate ROPs and alter $Ca^{2+}$ levels, which is required for pollen tube germination. We demonstrated that not all GEFs act redundantly during pollen germination, and we described for the first time a polar domain with spatiotemporal flexibility, which is crucial for the *de novo* establishment of a polar growth domain within a cell and, thus, for pollen function and fertilization success.

**Data availability statement:** All relevant data are within the paper and its Supporting Information files.

**Funding:** This work was supported by the Deutsche Forschungsgemeinschaft (DFG, https://www.dfg.de/) as a project to PD within the Collaborative Research Centre SFB924/3 (Project #170483403) and a project to PD within the DFG Priority Programme 2237 (Project #528090862). The funders had no role in study design, data collection and analysis, decision to publish, or preparation of the manuscript.

**Competing interests:** The authors have declared that no competing interests exist.

**Abbreviations:** ANX, ANXUR; BUPS, BUDDHA'S PAPER SEAL; CDS, coding sequence; CRIB, Cdc42- AND Rac-INTERACTIVE BINDING; iBAQ, intensity-based absolute quantifications; PAE, predicted aligned error; pLDDT, predicted local distance difference test; PRKs, POLLEN RECEPTOR KINASE; PRONE, plant-specific ROP nucleotide exchanger; RIC, ROP INTERACTING CRIB-CONTAINING PROTEIN; RIP, ROP INTERACTING PARTNER; RLKs, RECEPTOR-LIKE KINASES; ROP, RHO OF PLANTS; ROPGEFs, ROP GUANINE NUCLEOTIDE EXCHANGE FACTORS; TPM, transcripts per kilobase million.

## Introduction

Sexual reproduction is a fundamental and complex process in which male and female gametes fuse to form a zygote, which develops into an embryo. In Angiosperms, sperm cells have lost their motility, yet the male gametes must be delivered to the female gametes. This is achieved by a tip-growing pollen tube formed by the vegetative pollen cell, which encloses two sperm cells. This pollen tube grows from the papilla cells of the stigma on the flower surface into the transmitting tract toward the female gametophyte inside the ovary. After reaching the female gametophyte, the pollen tube ruptures and releases its enclosed sperm cells. In the last step of double fertilization, which is characterized by defined $Ca^{2+}$ signals in the female gametes, the sperm cells subsequently fuse with the egg cell and the central cell to form the zygote and the endosperm, respectively [1–5].

To protect the male gametophyte from environmental influences on its way to a compatible flower, the pollen is metabolically inactive, desiccated, and encapsulated in a thick and rigid pollen coat, forming the pollen grain. Once on the stigma of a compatible flower, the vegetative cell needs to be activated and polarize the tip growth machinery to a defined subcellular region to germinate from the pollen grain [6,7]. Pollen grains have apertures, areas in which the pollen coat is thinner, which predefine the possible emergence regions of the pollen tube in most angiosperms. However, in some species, such as *Arabidopsis thaliana,* the pollen tube emergence site is independent of these apertures and is predominantly defined by the contact site to the papilla cells. This requires sensing of the contact site and a growth machinery that can be polarized independently of the pollen morphology to loosen the pollen coat locally and allow the pollen tube's subsequent polar emergence [7]. Moreover, in most Angiosperms, the number of pollen grains exceeds the number of female gametes, causing competition between the individual pollen grains. Thus, the rapid establishment of cell polarity and polar growth initiation required for pollen germination is crucial in this competition and is decisive for fertilization success [1,8]. The factors responsible for sensing the papilla-pollen contact site and the polarization of the tip growth machinery in pollen grains are unknown. Moreover, the proteins required to transmit this polarization signal to the tip growth machinery are yet to be discovered. For this purpose, we investigated this crucial aspect of pollen germination.

During pollen tube tip growth, multiple receptor-like kinases (RLKs), like pollen receptor kinases (PRKs), Buddha's paper seal (BUPS), or Anxur (ANX) RLKs were shown to be required for pollen tube growth. BUPS and ANX RLKs are crucial for maintaining pollen tube integrity and preventing the rupture of germinated pollen tubes [9–11]. PRKs promote general pollen tube growth and are required for chemotaxis towards the female gametophyte [12–15]. PRKs have also been proposed to sense stigmatic signal peptides in Tomato and thus activate pollen germination, but no general germination-promoting function for these RLKs was shown [13,14,16]. Thus, it is still unknown whether PRKs play a general role in pollen activation and germination or which proteins are crucial for initiating pollen activation.

PLOS Biology

In various cell types and processes, RLKs activate Rho of plants (ROP) signaling pathways to establish cell polarity, promote polar growth, or confer immune responses [17–21]. ROP signaling pathways are mediated by plant-specific ROP GTPases, which are part of the Rho family of small GTP-binding proteins that act as molecular switches and cycle between an inactive GDP-bound state to an active GTP-bound state [22,23]. In their active state, ROPs interact with ROP interacting partner (RIP) and ROP interacting crib-containing protein (RIC) proteins, which facilitate the specific activation of downstream pathways that are required for polar growth [23–27]. During pollen tube tip growth, ROPs are essential for cell polarization and to promote tip growth [22,28]. Recently, it was shown that ROP signaling is additionally crucial for pollen germination, as the quadruple mutant *rop1;3;5;9* of all redundant, pollen-expressed ROPs is sterile and incapable of pollen germination [29]. As ROP signaling is required for pollen germination, we hypothesize that activators of ROP signaling are also crucial for pollen germination. However, it is unknown which ROP activators are required for pollen germination. The activation of ROPs is stimulated by ROP-specific guanine exchange factors (ROPGEFs), which facilitate the exchange from GDP to GTP. *Arabidopsis thaliana* has 14 of these ROPGEFs, hereafter called GEFs, which contain a conserved plant-specific ROP nucleotide exchanger (PRONE) domain and variable termini [30–33]. The PRONE domain forms a homodimer in which each protein binds a ROP protein and catalyzes the nucleotide exchange of the GTPase, as demonstrated by the structure of ROP4 together with the PRONE domain of GEF8 [34,35]. Individual GEFs exhibit specific expression patterns and functions in different cell types and polarity processes. We showed that GEF3, together with the previously known GEF4, is highly expressed in root hairs and promotes polarity establishment or root hair growth [17,36]. Specific GEFs establish the membrane domains required for xylem development, and we have shown that particular GEFs are expressed during phloem differentiation [37,38]. Additionally, to other processes, such as hormone signaling or pavement cell morphogenesis, GEFs were extensively studied in pollen tube tip growth [14,19,20,23,39,40].

Of the 14 GEFs of *Arabidopsis thaliana*, multiple is expressed in mature pollen grains, and four are reliably detected in transcriptomic and proteomic approaches (S1 Fig) [41]. Previous studies mainly focused on GEF12 and investigated its role in pollen tube growth and the activation of ROP signaling during tip growth. In addition to that, these studies showed that the activity of GEFs is promoted by the phosphorylation of their C-terminus by PRK RLKs or controlled by the cytosolic AGCVIII protein kinases AGC1.5 and AGC1.7, which phosphorylate the PRONE domain of GEFs. Overexpression or misregulation of GEFs results in a loss of polarity, pollen tube swelling, and unidirectional growth, while loss of GEF function leads to shorter pollen tubes [14,27,39,42–44]. As for ROPs, redundancy between multiple GEFs was indicated during pollen tube growth. A *gef1/gef9/gef12/gef14* quadruple mutant displayed a mild reduction in pollen tube length, but only two of the mutated *GEFs* are expressed in pollen [14]. Recently, a *gef8/gef9/gef11/gef12/gef13* quintuple mutant of all known pollen-expressed *GEFs* showed a decrease in pollen tube integrity during tip growth and reduced fertility [45]. These studies show the importance of GEFs in promoting and maintaining polar growth and, thus, male fertility. However, they only investigated pollen tube tip growth. The role of GEFs during pollen germination and which of these GEFs initiates and activates the growth process still need to be discovered.

Compared to the redundancy found among ROPs, GEFs can have specific functions during the establishment of a new polar domain and polar growth initiation. In root hairs, we showed that establishing the polar growth domain and tip growth are two processes activated by distinct GEFs. GEF3 is required to establish the root hair initiation domain and the polarization of ROP2, while GEF4 drives the subsequent tip growth [36]. In pollen tube germination, a polar growth domain must be established quickly from a dormant cell to provide an advantage over competing pollen. Moreover, in Arabidopsis, this polar protein accumulation must be spatially flexible, as the area of pollen tube germination is not predetermined and is defined by the region in contact with the papilla cells [2,7]. Therefore, pollen germination is a great model for studying the *de novo* formation of polar protein domains at the plasma membrane and understanding the spatiotemporal processes required to initiate polar growth. As GEF proteins have not been investigated during pollen germination, we investigated their role in this process to understand how GEFs activate ROP signaling in a spatiotemporally controlled manner to allow the *de novo* establishment of polar cellular growth.

We focused our research on the pollen-specific ROPGEFs (GEF8, GEF9, GEF11, GEF12, and GEF13) during pollen germination [41] and show that GEFs are distinctively crucial for pollen germination and male fertility, as mutants of *gef8*, *gef9* and *gef12* but not *gef11* influence pollen germination efficiency. We demonstrated that GEF8, GEF9, and GEF11, but not GEF12 or GEF13, form a transient polar domain at the plasma membrane of the pollen germination site and that these GEFs drive polar ROP activation and cellular growth. This novel subcellular localization highlights that GEFs have specific roles during cellular processes. Furthermore, this GEF accumulation appears in a biphasic temporal manner and can shift its location, which is the first description of a polar domain with such flexibility.

## Results

### GEF8, GEF9, and GEF11 biphasically accumulate at the pollen germination site

In flowering plants, pollen germination is essential for male fertility and, thus, successful double fertilization. Therefore, it is crucial to understand the molecular mechanisms activating pollen germination upstream of the essential ROP proteins [23,29]. Even though multiple GEFs were shown to be involved in pollen tube growth, the GEFs involved in ROP activation during pollen germination still need to be determined [14,23,45]. To investigate the role of GEFs during pollen tube germination, we focused on five GEF proteins (GEF8, GEF9, GEF11, GEF12, and GEF13) that had been found to be expressed in mature pollen (S1 Fig) [41]. According to phylogenetic analysis, these five GEFs belong to one branch that can be subdivided into two clades that contain GEF8 and GEF9 on the one side and GEF11, GEF12, and GEF13 on the other clade (S1 Fig). This phylogeny is in line with previous analysis and shows that these five Arabidopsis pollen-expressed GEFs are closely related [32,46]. We fused these five GEFs N-terminally with the yellow fluorescent protein mCitrine (mCit) under the control of their endogenous promoter fragments to confirm their expression. We found mCit-GEF8, mCit-GEF9, mCit-GEF11, and mCit-GEF12 signals in mature pollen, but no signal was observed for mCit-GEF13 (Figs 1A and S2). We assessed mCit-GEF localization by live-cell imaging of germinating pollen grains *in vitro* on pollen germination medium with 24-epibrassinolides (PGM) [47,48]. Shortly after imbibition on PGM, all GEFs showed a similar localization and were evenly distributed in the cytoplasm (Figs 1A and S2 and S1–S5 Videos). After several minutes, we observed that mCit-GEF8, mCit-GEF9, and mCit-GEF11 accumulated at a defined region of 3–6 μm at the cell periphery, which was always the future germination site. Such an accumulation was not observed for mCit-GEF12 or mCit-GEF13 expressed under the control of a *GEF12* promoter fragment (Figs 1A and S2 and S1–S5 Videos). The accumulation of mCit-GEF8, mCit-GEF9, and mCit-GEF11 indicated a specific function of these GEFs during pollen germination initiation. To further characterize this behavior, we quantitatively compared the timing, strength, and frequency of protein accumulation. We defined the first frame of a visibly emerged pollen tube as a reference time point 0, with negative time points before and positive time points after this reference point.

We used kymographs along a line across the pollen grain and around the pollen grain to visualize protein localization over time and found that the observed accumulation is flexible in timing, length, and, to some degree, location (Fig 1B). To quantify these differences, we measured the signal intensity of mCit-GEF8, mCit-GEF9, mCit-GEF11, and mCit-GEF12 at the pollen germination site in relation to the remaining pollen grain (Fig 1C–1G and S1 Data). This showed that mCit-GEF12 was mostly nonpolar and only accumulated slightly in 2 out of 17 cases (11.8%, Fig 1A–1G). Therefore, we used the non-accumulating mCit-GEF12 as a reference protein to generate a threshold that allowed us to determine protein accumulation for other proteins. Quantifying the polarization of mCit-GEF8, mCit-GEF9, and mCit-GEF11 showed that they have a similar polarization strength, with mCit-GEF8 displaying the highest overall values (Fig 1F). Moreover, all three GEFs accumulate in most pollen grains with a similar frequency (Fig 1G, mCit-GEF8: 90.9%, mCit-GEF9: 85.7%, mCit-GEF11: 88.2%). However, the accumulation time, duration, and persistence were variable between individual pollen grains of the same construct and between different GEFs, indicating that this process is flexible and does not depend on strict timing (Fig 1C–1E). Similarities are that mCit-GEF8, mCit-GEF9, and mCit-GEF11 all transiently accumulated at the cell periphery. The accumulations lasted between

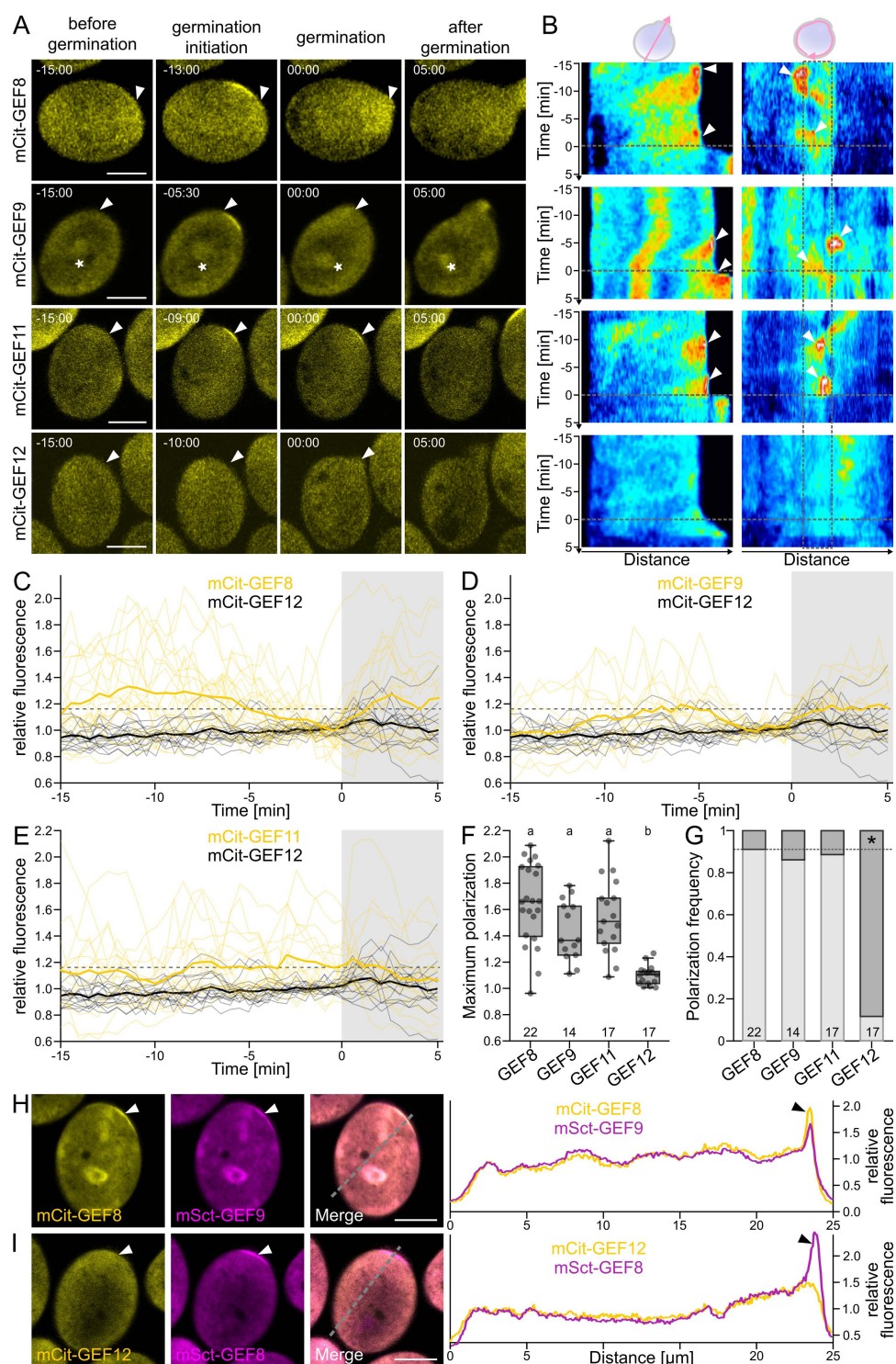

**Fig 1. GEF8, GEF9, and GEF11 specifically accumulate at the pollen germination site before germination initiation. (A)** Localization of mCit-GEF8, mCit-GEF9, mCit-GEF11, and mCit-GEF12 under their respective promoters during pollen germination. Timepoint 0 corresponds to the beginning

of pollen tube emergence, arrowheads mark the site of pollen emergence, and asterisks mark mCit-GEF9 localization around the sperm cells. **(B)** Kymographs of time-lapse images corresponding to **(A)** along a line crossing the pollen through the germination site (left) or around the pollen grain (right). The dotted line indicates time point 0 of pollen tube emergence; the dotted line box in the right kymographs encloses the pollen germination site; arrowheads highlight protein accumulations at the pollen germination site. **(C–E)** Relative fluorescence intensity profiles at the pollen germination site of mCit-GEF8 (yellow, $n = 22$), mCit-GEF9 (yellow, $n = 14$), and mCit-GEF11 (yellow, $n = 17$), all in comparison to mCit-GEF12 (black, $n = 17$). Thin lines show individual measurements, and thick lines represent the average of all samples. **(F, G)** Quantification of measurements shown in **C–E**. **(F)** Maximum polarization value per pollen grain before germination. Letters above plots show significantly different groups according to a one-way ANOVA with Tukey's test ($p < 0.05$). **(G)** Portion of pollen with a polarized signal (polarization frequency) before germination. The asterisk indicates a significant difference according to a $X^2$-test ($p < 0.05$). **(H, I)** Colocalization (left) and intensity profiles along a line, as indicated in the merged image, across the pollen grain through the pollen germination site (right) of mCit-GEF8 with mSct-GEF9 **(E)** and mCit-GEF12 with mSct-GEF9 **(F)** expressed under their respective promoters. All scale bars represent 10 μm. For underlying data of all quantification see S1 Data.

2 min and 12 min and occurred within 15 min before pollen germination. In addition, the accumulation is lost after several minutes for all three GEFs but reappears at the pollen germination site when growth is starting, leading to a biphasic accumulation. We hypothesize that the first accumulation marks the initiation of the germination process, while the second accumulation happens during the start of tip growth. Besides these similarities, we found that, on average, mCit-GEF8 accumulates stronger (1.64×) and longer (7.0 min) than mCit-GEF9 (1.44×, 4.8 min) or mCit-GEF11 (1.54×, 5.3 min) (Fig 1F). Additionally, the timing of accumulation differs among GEF8, GEF9 and GEF11. Even though all three GEFs are variable and no clear time point of accumulation can be determined, the timing of mCit-GEF8 and mCit-GEF9 accumulations are more clustered and less polar within the 3 min before germination than those of mCit-GEF11, which leads to a more pronounced average accumulation graph for mCit-GEF8 and mCit-GEF9 (Fig 1C–1E). We found that the average of mCit-GEF8 accumulations has its maximum 11.5 min before germination, while the average maximum for mCit-GEF9 is 6.0 min and for mCit-GEF11, 3.0 min before germination (Fig 1C–1E). Together, these results show that GEF8, GEF9, and GEF11 accumulate at the pollen germination site but have different timing and variability of accumulation, which indicates differences in their cell biological function.

Interestingly, we observed that the accumulation of mCit-GEF8/mCit-GEF9 was sometimes slightly shifted laterally compared to the final germination site (Fig 1A and 1B and S1 and S3 Videos). This was observed in a similar frequency of approximately 50% for mCit-GEF8, mCit-GEF9, and mCit-GEF11. However, in all cases, both sites were always near each other, and the 3–6 μm accumulations shifted within 5–10 μm around the center of the germination site. Kymographs around the pollen grain showed that the accumulations drifted laterally and could be followed over time to their final destination (Fig 1B, GEF8 and GEF11). This indicates a certain positional flexibility during the assembly of all required proteins of the tip growth machinery. Such flexibility of the polar growth domain is crucial for pollen function, especially in species like *Arabidopsis thaliana*, in which the pollen apertures do not predetermine the pollen germination site [7]. To confirm the accumulation and timing of GEF8 and GEF9 compared to the evenly distributed GEF12, we simultaneously observed mCit-GEF8 with mScarlet-GEF9 (mSct-GEF9) or mSct-GEF8 with mCit-GEF12. mCit-GEF8 and mSct-GEF9 both accumulated with a similar timing at the same location, confirming the observations in the single marker lines. However, mCit-GEF12 was still evenly diffused in the cytoplasm, with no significant accumulation when a clear accumulation of mSct-GEF8 was visible (Fig 1H and 1I and S1 Data). These results show differences between individual GEFs, as mCit-GEF12 and mCit-GEF13 do not show specific accumulations during pollen germination, while mCit-GEF8, mCit-GEF9, and mCit-GEF11 specifically accumulate at the pollen germination site. Moreover, mCit-GEF8, mCit-GEF9, and mCit-GEF11 show a similar biphasic accumulation at the pollen germination site, with large variability between individual pollen grains, but also temporal differences between the average accumulation of the different proteins. Taken together, the differential localization during pollen germination indicates a specific and subcellular localized role for GEF8, GEF9, and GEF11 in initiating pollen germination.

## Distinct GEFs are required for pollen germination *in vitro*

In light of the specific accumulation of GEFs during pollen germination, we used loss-of-function mutant lines to investigate the function of GEFs in pollen germination. We used available T-DNA lines for *gef9-t1* (GK-717A10), *gef11-t1*

*(SALK_126725), and gef12-t1* (SALK_103614) and confirmed that they do not produce full-length mRNA (S3 Fig). Additionally, we generated CRISPR-Cas9 full-length deletion mutants, resulting in *gef8-cΔ1, gef8-cΔ2, gef9-cΔ1, gef11-cΔ1, gef12-cΔ1* single mutants (S3 Fig). By crossing these lines, we generated several double and triple mutants. Moreover, we generated *gef8c-Δ3;9-cΔ2* double mutants by CRISPR-Cas9 in a single line, and triple mutants of *gef8-cΔ3;9-t1;11-cΔ2* and *gef8-cΔ3;9-cΔ2;12-cΔ2* by mutating *GEF11* or *GEF12* in the according *gef8/gef9* mutant background (S3 Fig). Col-0 was used as a wild-type reference and reached a pollen germination efficiency of 87% 4 h after imbibition on PGM (Fig 2A and 2B and S1 Data). Both alleles of *gef11-t1* (86%) and *gef11-cΔ1* (90%) showed germination efficiencies similar to Col-0, while *gef9-t1* (79%)*, gef12-t1* (70%) and *gef12-cΔ1* (78%) were slightly lower but not consistently different from Col-0. In comparison to *gef9-t1*, *gef9-cΔ1* (55%) showed a significant reduction of pollen germination efficiency, indicating partial remaining GEF9 function in *gef9-t1*, which could be explained by the location of the T-DNA insertion in the fifth intron, even though no full-length mRNA in *gef9-t1* was detectable by RT-PCR on flower cDNA (S3 Fig). The germination efficiency of *gef8-cΔ1* (63%) and *gef8-cΔ2* (64%) were comparable to *gef9-cΔ1* and were significantly different from Col-0. We were able to rescue *gef8-cΔ1* with the *GEF8p*::mCit-GEF8 construct, which led to a germination efficiency of 83%, proving that the absence of GEF8 caused the phenotype and confirmed the functionality of the mCit-fusion constructs (Fig 2A). The pollen germination was more severely reduced in *gef8-cΔ1;9-t1* (36%) and *gef8-cΔ3;9-cΔ2* (37%) double mutants, while *gef8-cΔ1;12-t1* (70%) and *gef9-t1;12-t1* (66%) double mutants displayed no significant reduction of germination efficiency in comparison to the *gef8* and *gef9* single mutants. Considering the similar localization, it is surprising that gef8-cΔ1;11-cΔ1 (84%) and gef9-t1;11-t1 (84%) double mutants did not enhance the phenotype but rescued the effect of the *gef8* and *gef9* single mutants with wild-type like pollen germination, indicating opposite functions of GEF8/9 and GEF11. As we still observed significant pollen germination in *gef8;gef9* double mutants, we suspected remaining redundant GEF function. Therefore, we analyzed triple mutant combinations of *gef8, gef9, gef11, and gef12*. In *gef9-t1;11-t1,12-t1* (78%) and *gef8-cΔ1;9-t1;11-cΔ2* (24%) triple mutants, pollen germination efficiency was not different to *gef9-t1;12-t1* or *gef8-cΔ1;9-t1* double mutants, confirming that GEF11 is not required for pollen germination. In contrast, *in vitro* pollen germination was almost completely abolished in *gef8-cΔ1;9-t1;12-t1* (2%) and *gef8-cΔ2;9-cΔ2;12-cΔ2* (1%), showing that these three GEFs are essential for pollen germination (Fig 2A). In summary, this showed that GEF11 is not required for pollen germination in *Arabidopsis thaliana*, and it might even have opposing effects to *GEF8*, *GEF9, and GEF12*. GEF8 and GEF9 play a major role in pollen germination, as they both display a significant deficiency as single mutants, and this effect is amplified in the double mutant. The normal pollen germination efficiency of GEF12 single mutants shows a minor function of GEF12 in this process. However, because pollen germination was only abolished in the combination of *gef8*, *gef9*, and *gef12*, a genetic redundancy between these three GEFs can be assumed, while they are not redundant to *GEF11*.

## GEF function is crucial for efficient male fertility *in vivo*

After we showed the necessity of GEFs for pollen germination *in vitro*, we assessed their role in pollination and male fertility *in vivo.* To determine the transmission efficiency of *gef* mutant alleles, we performed reciprocal crosses using Col-0 with *gef8, gef9, and gef12* mutant combination lines, in which one *gef* allele was heterozygous, leading to an expected transmission of 50% for this *gef* allele in the F1 offspring (Figs 2C and S4 and S1 Data). In the control direction with *gef* mutant lines as female, pollinated with Col-0 pollen, we did not find any significant reduction in the transmission of the mutant alleles, showing that *gef8*, *gef9*, and *gef12* are not required for female fertility (S4 Fig). When using *gef* mutant lines for pollination, we did not observe significant reductions from the expected transmission *in gef8, gef9, and gef12* single and double mutant combinations (Figs 2C and S4). However, both *gef8-cΔ1;9-t1±;12-t1* and *gef8-cΔ2;9-cΔ2;12-cΔ2±* triple mutant had a significantly reduced transmission of the *gef9-t1* allele (35%) and the *gef12-cΔ2* allele (23%), respectively (Figs 2C and S4). This effect on male fertility was lower than expected from the *in vitro* experiments but confirmed that *GEF8, GEF9, and GEF12* are crucial for male fertility, likely by promoting efficient pollen germination.

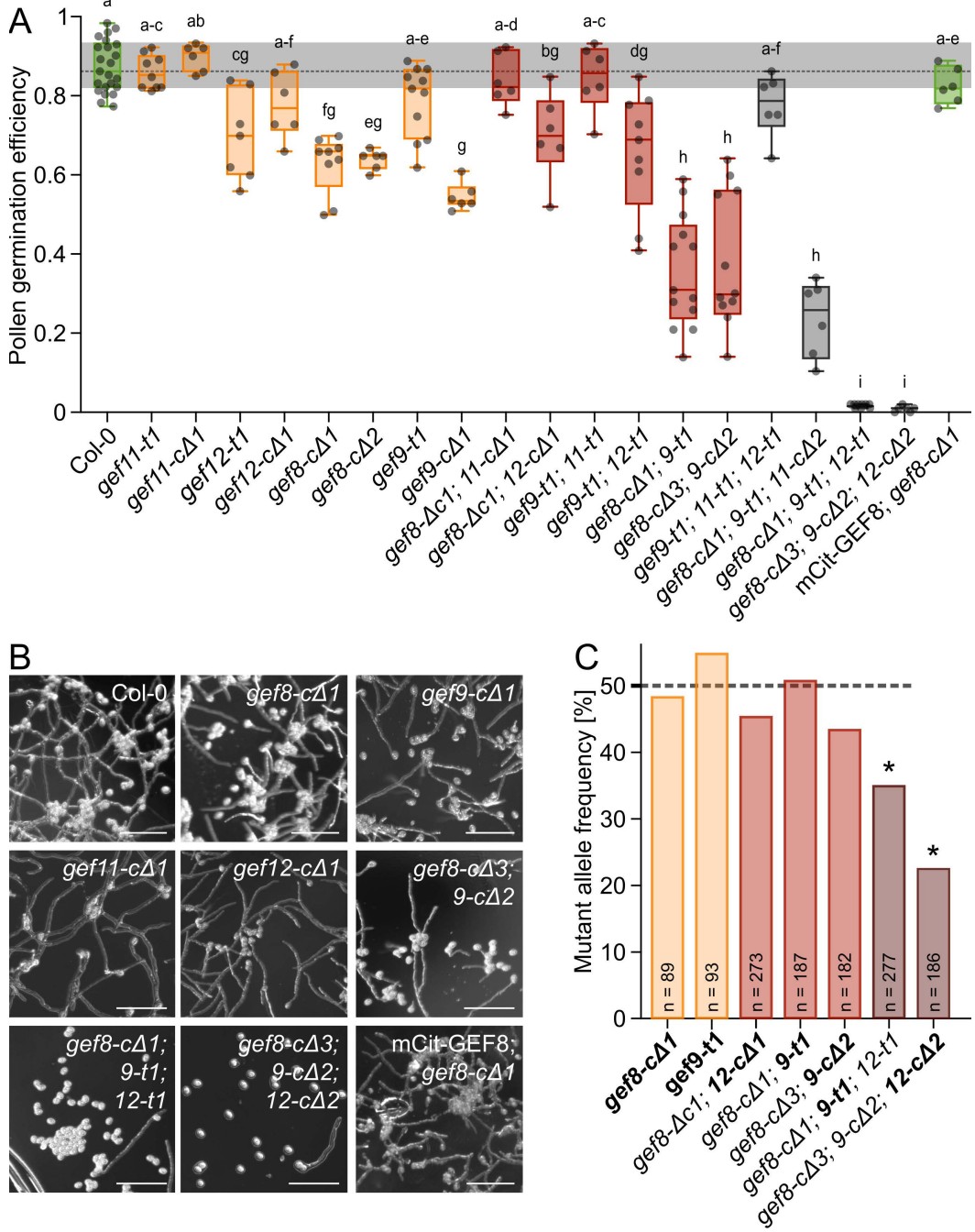

**Fig 2. GEF8, GEF9, and GEF12 are necessary for pollen germination and male fertility. (A)** Pollen germination efficiency of *in vitro* germinated pollen 4 h after imbibition on pollen germination media. Each point represents one replicate with more than 150 pollen grains. Groups of statistically significant differences according to a one-way ANOVA with Tukey's test ($p < 0.05$) are indicated with letters. **(B)** Representative images of *in vitro* germinated pollen 4 h after imbibition on PGM. **(C)** Quantification of mutant allele frequency in F1 generation of reciprocal crosses with Col-0 as female and the indicated mutants as pollen donors. The heterozygous allele of each genotype is indicated in bold. Asterisks indicate a significant difference in the allele frequency from the expected 50% according to $X^2$-test ($p < 0.05$). For underlying data of all quantification see S1 Data.

## The C-terminal domain is necessary and sufficient for GEF8 and GEF9 accumulation

We showed that GEF8, GEF9, and GEF11 have distinct localizations compared to GEF12 and that *GEF8, GEF9,* and *GEF12* are crucial for efficient pollen germination. Therefore, we focused our investigations on these proteins to understand their differences in localization and function. To investigate which GEF proteins' components are responsible for the specific accumulation of GEF8 and GEF9, we analyzed different domains of GEFs during pollen germination. GEFs consist of a conserved catalytic PRONE domain and variable N- and C-terminal regions. The existing crystal structures of the PRONE domain of GEF8, together with ROP4, showed that these proteins form a tetramer in which two GEFs dimerize, each having one ROP bound (Fig 3A and 3B) [34]. However, the terminal regions of GEF8 were not represented in this structure, as they are thought to be intrinsically disordered. Protein structure predictions of full-length GEF8 and ROP1 as separate proteins using AlphaFold2 and matched on the existing PRONE8-ROP4 structure (RCSB-PDB: 2NTY) with ChimeraX confirmed that these terminal regions are primarily unstructured and mostly disordered, with only small helical elements (Figs 3A and S5) [34,49,50]. Even though the prediction of the terminal regions has very low confidence, all five models have in common that the N- and C-terminus fold around the complex (Figs 3A and S5), indicating potential regulatory functions as it was described previously [39,51]. We made several mutant and deletion constructs to understand the function of GEF8 and GEF9 during pollen germination and unravel the protein features responsible for their specific biphasic accumulation (Fig 3B and 3C). We started by deleting the variable N-terminus of GEF8 (mCit-GEF8$^{\Delta N}$) and GEF9 (mCit-GEF9$^{\Delta N}$). The N-terminal deletion did not abolish the specific localization of either protein, as we could still observe an accumulation at the germination site before pollen tube emergence in 70% and 75%, respectively (Figs 3C, 3G and S6 and S8 Videos and S1 Data). However, the polarity strength was significantly reduced, with an average maximum polarization of 1.33× for mCit-GEF8 and 1.34× for mCit-GEF9, compared to 1.64× and 1.44× in both wild-type proteins (Figs 1, 3C, 3F and S6). This remaining but reduced polarity was also reflected in the functionality of the mCit-GEF8$^{\Delta N}$ truncation construct, which could partially rescue the loss of pollen germination efficiency in *gef8-cΔ1* pollen (Fig 3H). More severe effects were observed in constructs in which the variable C-terminal region of GEF8 (mCit-GEF8$^{\Delta C}$) and GEF9 (mCit-GEF9$^{\Delta C}$) was deleted. mCit-GEF8$^{\Delta C}$ and mCit-GEF9$^{\Delta C}$ were cytosolic and only showed membrane attachment during pollen germination in 15% and 14% of the observed pollen, which is similar to the nonpolar mCit-GEF12. This loss of polarity can also be seen by their average maximum polarization, which is reduced to 1.18× for GEF8 and 1.13× for GEF9 (Figs 3C, 3F-3G and S6 and S7 and S9 Videos). Moreover, the mCit-GEF8$^{\Delta C}$ truncation construct had no functionality as it had a comparable pollen germination efficiency to *gef8-cΔ1* pollen (Fig 3H). The loss of accumulation and functionality of mCit-GEF8$^{\Delta C}$ and mCit-GEF9$^{\Delta C}$ showed that the GEF C-terminus is necessary for membrane attachment, and the accumulation of GEF8 and GEF9 is required during pollen germination. Because the C-terminal domain is required for the accumulation of GEF8/GEF9 and such accumulation is not observed for mCit-GEF12 throughout the pollen germination process, we swapped the C-terminal domain of GEF12 with those of either GEF8 (mCit-GEF12$^{GEF8C}$) or GEF9 (mCit-GEF12$^{GEF9C}$) (Fig 3D and S10-S11 Videos). mCit-GEF12$^{GEF8C}$ and mCit-GEF12$^{GEF9C}$ both accumulated at the pollen germination site before pollen tube emergence in 85% and 87% of the observed cases, with an average maximum polarization of 1.50× for m-Cit-GEF12$^{GEF8C}$ and 1.49× for mCit-GEF12$^{GEF9C}$, both comparable to the mCit-GEF8 and mCit-GEF9 and significantly different from wild-type mCit-GEF12 (Figs 3D, 3F-3G and S6 and S1 Data). In line with these results, mCit-GEF12$^{GEF8C}$ was capable of partially rescuing the loss of pollen germination efficiency in *gef8-cΔ1* pollen. However, the expression of mCit-GEF12 in *gef8-cΔ1* also partially rescued this pollen germination phenotype (Fig 3H). Together, this showed that the C-terminal domains of GEF8 and GEF9 are necessary for their polar accumulation. However, this accumulation is only required for the functionality of GEF8, while cytosolic GEF12 is also capable of rescuing the loss of *GEF8*. These distinct effects of the C-terminus on GEF localization and the redundancy in their functionality confirm the genetic experiments in higher-order mutants (Fig 2). The domain-swap constructs, fusing the C-terminus of GEF8 or GEF9 to GEF12, showed that both C-termini are sufficient to recruit other GEFs to the cell periphery of the future pollen germination site. However, the GEF8 C-terminus alone was not capable of polarizing during pollen germination (S6 Fig and

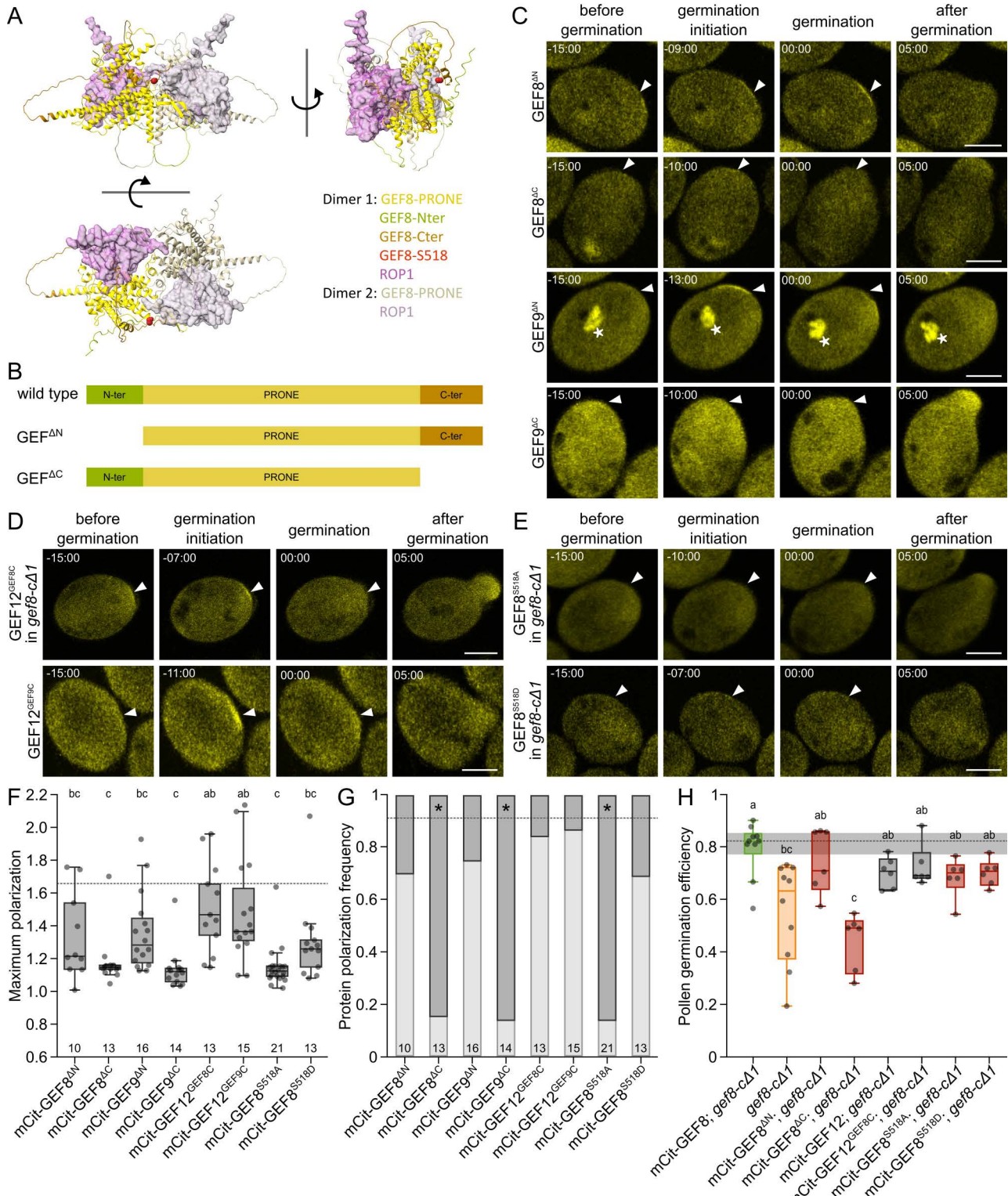

**Fig 3. The C-terminus of GEF8 and GEF9 is necessary and sufficient for protein accumulation. (A)** AlphaFold2 protein structure prediction of full-length GEF8 and ROP1 (Rank 1, predicted separately), matched on the PRONE8-ROP4 double-dimer structure (PDB: 2NTY). Different angles are shown, and the terminal regions are indicated by different colors. **(B)** Schemes of GEF protein structure and investigated truncation constructs, with

variable N and C terminal domains indicated as in **(A)**. **(C–E)** Protein localization of different mutant constructs during pollen germination. Timepoint 0 corresponds to the beginning of pollen tube emergence, arrowheads mark the site of pollen emergence, and asterisks mark localization around the sperm cells. **(C)** Truncation constructs mCit-GEF8$^{\Delta N}$, mCit-GEF8$^{\Delta C}$, mCit-GEF9$^{\Delta N}$, and mCit-GEF9$^{\Delta C}$ in Col-0. **(D)** Domain swap constructs of GEF12 with alternative C-terminal domain, mCit-GEF12$^{GEF8C}$ in *gef8-cΔ1* and mCit-GEF12$^{GEF9C}$ in Col-0. **(E)** Phosphorylation site mutations of GEF8-S518 to a phospho-mimic (mCit-GEF8$^{S518D}$) and phospho-dead variant (mCit-GEF8$^{S518A}$) variant expressed in *gef8-cΔ1*. All scale bars represent 10 µm. **(F, G)** Quantification of measurements shown in **C–E**. **(F)** Maximum polarization value per pollen grain before germination. Letters above plots show significantly different groups according to a one-way ANOVA with Tukey's test ($p < 0.05$) with mCit-GEF8 (dotted line, group a) as a reference. **(G)** Portion of pollen with a polarized signal (polarization frequency) before germination. The dotted line indicates mCit-GEF8 frequency. The asterisks indicate a significant difference to mCit-GEF8 or mCit-GEF9 according to a X$^2$-test ($p < 0.05$). **(H)** Pollen germination efficiency of *in vitro* germinated pollen 4 h after imbibition on pollen germination media. Each point represents one replicate with more than 50 pollen grains. Groups of statistically significant differences according to a one-way ANOVA with Tukey's test ($p < 0.05$) are indicated with letters. For underlying data of all quantification see S1 Data.

S12 Video), indicating that the context of a functional PRONE domain is required for the polarization ability of the C-terminal domain of GEF8 and GEF9.

## A phosphorylation site in the C-terminus of GEFs influences their accumulation

To further understand the mechanism of GEF activation and localization, we investigated the role of phosphorylation in GEF8 and GEF9 C-terminal domains. The C-terminus of GEFs was shown to be phosphorylated by RLKs at a serine of the conserved SPxxRH motif, leading to the activation of the PRONE domain [20,39,52]. The phosphorylation at this site was recently confirmed by proteomic approaches [41]. To elucidate any potential functional effect of the phosphorylation of this serine on pollen germination, we mutated this serine to a phospho-dead (S518A) and potentially phospho-mimic (S518D) versions of GEF8, transformed them into the *gef8-cΔ1* background, and observed these variants during pollen germination (Figs 3E and S6 and S13-S14 Videos). Quantifications showed that GEF8$^{S518A}$ and GEF8$^{S518D}$ displayed a significant loss of polarization at the pollen tube germination site (Figs 3E-3G and S6 and S1 Data). However, the polarization of GEF8$^{S518D}$ (1.30×) was still slightly larger than the GEF12-like GEF8$^{S518A}$ (1.15×) (Fig 3F). This difference leads to a significant loss of polarization frequency for GEF8$^{S518A}$ (14%), which is similar to GEF12, while the phospho-mimic GEF8$^{S518D}$ still remains a polarization frequency of 69%, even though these accumulations are significantly reduced compared to wild-type GEF8 (Fig 3G). Interestingly, GEF8$^{S518A}$ and GEF8$^{S518D}$ both rescued the *gef8-cΔ1* phenotype to a similar degree (Fig 3H). This indicates that the phosphorylation of the C-terminus influences protein accumulation and polarization, but not the functionality of GEF8, showing that other parts of the C-terminal domain are required to regulate GEF function. Moreover, as both mutants show slightly different polarization defects, we hypothesize that protein polarization does not depend on the phosphorylation status, but rather, the phosphorylation reaction by RLKs itself is critical for GEF accumulation. In summary, the deletion constructs of GEF8 and GEF9, in combination with the domain swap experiments, show that the C-terminal domain is necessary for the distinct accumulation of GEF8 and GEF9 and sufficient to transfer this function onto GEF12. In addition, phosphorylation of GEF8$^{S518}$ has no influence on protein functionality, but it plays an essential role in GEF8 accumulation at the germination site.

## GEF8 is necessary for ROP activation

GEFs activate ROPs by exchanging GDP for GTP, leading to the recruitment of RIP or RIC proteins and the subsequent activation of downstream processes [21,23]. For example, RIC3 and RIC4 were shown to regulate pollen growth by mediating the modulation of Ca$^{2+}$ fluxes and F-actin formation [53]. RICs contain a CDC42/RAC INTERACTIVE BINDING (CRIB) motif, which mediates the interaction with active GTP-bound ROPs. ROP1, ROP3, and ROP5 are specifically expressed in pollen grains and are essential for pollen germination [29]. However, the localization of these proteins has so far not been reported in detail during pollen germination. We observed mCit-ROP1, mCit-ROP3, and mCit-ROP5 under their endogenous promoters, and all three ROPs localized to the cytoplasm without

any specific accumulation at the site of germination in more than 10 pollen grains each (Fig 4A and 4B and S15–S17 Videos). Even though there was a moderate increase in signal intensity for all ROPs in the cytosolic region of the pollen germination site, no membrane association or defined accumulation at the cell periphery as found for GEF8 or GEF9 could be observed (Fig 4A–4B). Therefore, we did not focus on ROPs in our further analysis, but on their activity and downstream ROP-signaling events. To polarly activate ROP signaling, ROPs do not necessarily need to polarize themselves if they are evenly distributed and locally activated. To identify such polarized ROP activation, we utilize the CRIB domain, which is sufficient for active ROP binding, as a biosensor for the localization of active ROPs [54]. We used the CRIB motif of RIC4 (CRIB4) fused to mCitrine under the control of the *GEF12* promoter fragment (*GEF12p*::CRIB4-mCit) as an indicator of ROP activity during pollen germination (Fig 4C and 4D and S18 Video). CRIB4-mCit accumulated at the cell periphery of the pollen germination site before germination in all 12 cases (Figs 4C–4E and S7 and S18 Video and S1 Data). The timing of the initiation of CRIB4-mCit accumulation was similar to that observed for GEFs, within 15 min before germination and an average maximum polarization 8.5 min before germination. In addition, the overall width of the accumulation of CRIB4-mCit was similar to that of mCit-GEF8 and mCit-GEF9 (3−6 µm). However, compared to GEFs, the accumulation of CRIB4-mCit was more persistent at the pollen germination site, with an average polarization time of 13.7 min. In 4 of 12 cases, the accumulation persisted continuously until germination, indicating that ROP activity can be maintained even if GEFs do not persist at the pollen germination site. To show the causality between the accumulation of GEFs and CRIB4, we investigated CRIB4-mCit localization in a *gef8-cΔ1* background. In *gef8-cΔ1*, CRIB4-mCit still accumulated in 83% of the cases (Figs 4C–4E and S7 and S19 Video). However, the accumulations of CRIB4-mCit in *gef8-cΔ1* were significantly less polar, with an average maximum polarization of 1.30× compared to CRIB4-mCit in Col-0 of 1.59× (Figs 4F and S7). This reduction in polarization was, in most cases, just above the polarization threshold and was therefore not obvious without quantification (S7 Fig). These results suggest that ROPs are activated polarly at the germination site during the germination process with a similar timing to GEF accumulation, and this local ROP activation depends on the activity of GEF8, which shows a functional link between GEF8/9 accumulation and ROP activation.

## GEF8 oscillation leads to changes in calcium oscillation

ROP signaling leads to the activation of downstream pathways required for pollen tube growth, such as $Ca^{2+}$ oscillations, which are essential for pollen tube growth and guidance [23,52,55]. $Ca^{2+}$ undergoes fine-tuned oscillations and concentration variations to regulate polar growth [56]. Furthermore, $Ca^{2+}$ fluctuations are described during pollen germination *in vivo*, but their timing to other activators of pollen germination and their cause during pollen activation still need to be understood [57]. We used the $Ca^{2+}$-indicator RGeco1 under the control of the *Lat52* promoter to monitor $Ca^{2+}$ levels during pollen germination in Col-0 (Figs 4G, 4H and S7 and S20 Video and S1 Data). Compared to the regular short oscillations during pollen tube growth, we observed long $Ca^{2+}$ elevations that lasted several minutes in all 13 cases. These $Ca^{2+}$ elevations appeared at all times before pollen germination, and no specific onset time point could be defined (S7 Fig). To investigate the association between GEF function and long $Ca^{2+}$ elevations, we observed RGeco1 in *gef8-cΔ1* during pollen germination (Figs 4G, 4H and S7 and S21 Video). Interestingly, the $Ca^{2+}$ elevation pattern was not abolished but differed from the pattern observed in Col-0. Compared to Col-0, the RGeco1 signal in the *gef8-cΔ1* mutant did not increase as often but rather showed lower intensity peaks throughout the germination process. We observed 4–7 (average 5.3) such strong $Ca^{2+}$ elevations in Col-0, while we only observed 2–5 (average 3.6) strong $Ca^{2+}$ elevations in *gef8-cΔ1* mutant pollen (Figs 4G–4I and S7 and S1 Data). This result further emphasizes the necessity of GEF8 for ROP activation, resulting in normal $Ca^{2+}$ elevation patterns and pollen germination, which is not abolished but altered in pollen grains lacking GEF8 activity. In summary, the absence of GEF8 impacted CRIB4-mCit (active ROP biosensor) and $Ca^{2+}$ fluctuation patterns, showing the crucial function of GEFs in activating and fine-tuning ROP signaling, leading to pollen germination.

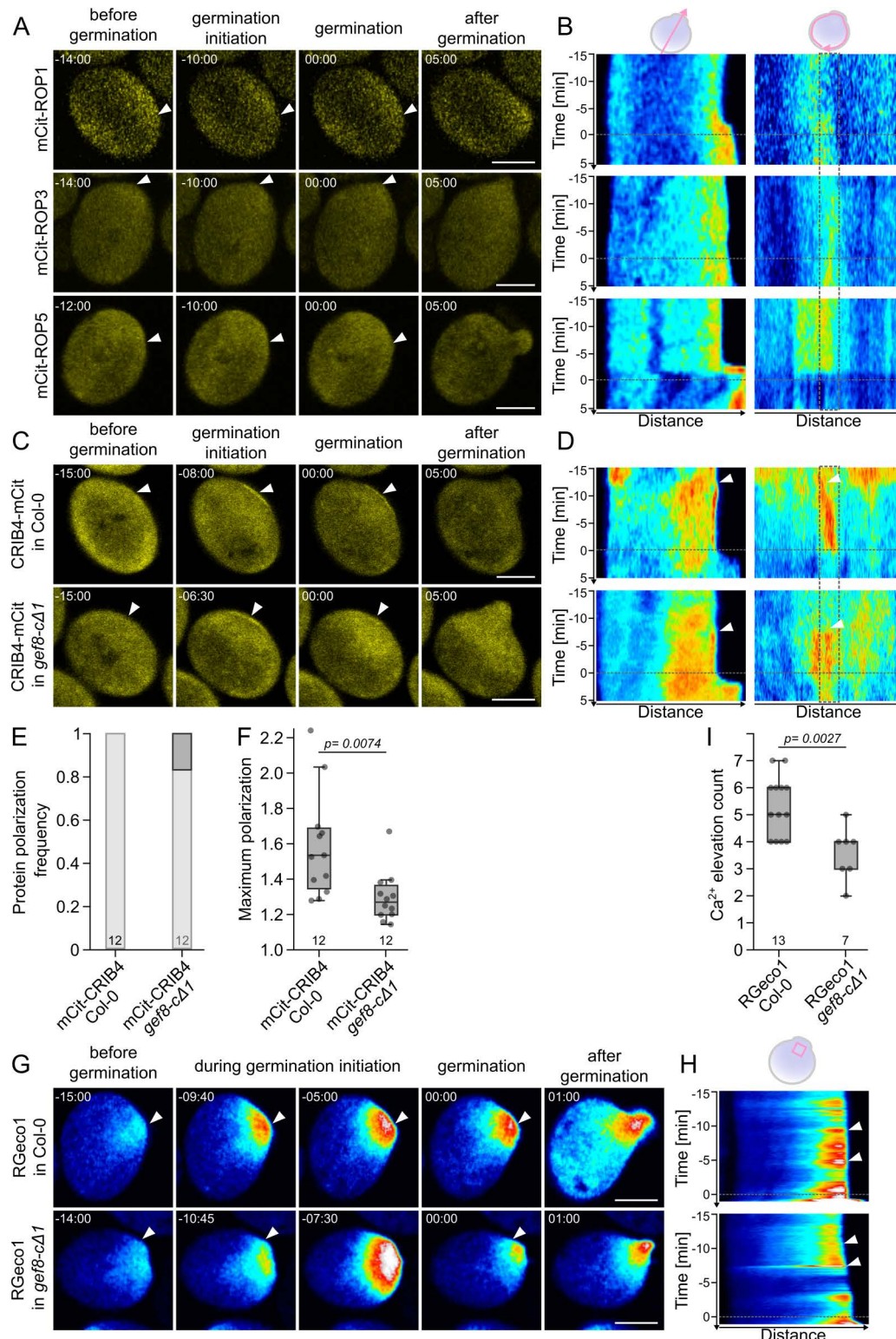

**Fig 4. GEF8 is required for polar ROP activation and Ca²⁺ signaling. (A)** Localization of mCit-ROP1, mCit-ROP3, and mCit-ROP5 under their respective promoter during pollen germination. Timepoint 0 corresponds to the beginning of pollen tube emergence, and arrowheads mark the site of pollen emergence. **(B)** Kymographs of time-laps images corresponding to **(A)** along a line crossing the pollen through the germination site (left) or around the

pollen grain (right). The dotted line indicates time point 0 of pollen tube emergence; the dotted box in the right kymographs encloses the pollen germination site; arrowheads highlight protein accumulations at the pollen germination site. **(C)** Localization of the ROP activity indicator *GEF12p*::CRIB4-mCit in Col-0 and *gef8-cΔ1* backgrounds. **(D)** Kymographs of time-laps images corresponding to **(C)** along a line crossing the pollen through the germination site (left) or around the pollen grain (right). Arrowheads highlight protein accumulations at the pollen germination site. **(E, F)** Quantification of measurements in germinating pollen grains expressing *GEF12p*::CRIB4-mCit in Col-0 (*n* = 12) and *gef8-cΔ1* (*n* = 12). **(E)** Portion of pollen with a polarized signal (polarization frequency) before germination. No significant difference was found according to a $X^2$-test ($p < 0.05$). **(F)** Maximum polarization value per pollen grain before germination. *p*-values of a Student *t* test is shown. **(G)** *LAT52p*::RGeco1 Ca$^{2+}$ biosensor in Col-0 and *gef8-cΔ1* background. **(H)** Kymographs of time-laps images corresponding to **(E)** along a line crossing the pollen through the germination site. Arrowhead highlights signal increases at the pollen germination site during germination initiation, which are represented by images. **(I)** Quantification of the number of detected large Ca$^{2+}$ elevations *p*-values of a Student *t*-test is shown. All scale bars represent 10 μm. For underlying data of all quantification see S1 Data.

## Discussion

Pollen germination is a critical step in plant fertilization, and characterizing the underlying protein functions is crucial for understanding the activation of this process. Here, we identified specific ROPGEFs required for pollen germination and male fertility. These GEFs have distinct functions during pollen germination, which are conferred, in parts, by their C-terminal domain. Five of the 14 GEFs are expressed in mature Arabidopsis pollen (S1 Fig). We confirmed the presence of GEF8, GEF9, GEF11, and GEF12 using translational fusion lines but not GEF13 (Figs 1 and S2). As recent proteomics analysis only found very low amounts of GEF13 in mature pollen, it remains unclear whether GEF13 is irrelevant for pollen tube growth or is very specifically translated [41]. Previous results also indicated the presence of GEF1 and GEF14 during pollen tube growth, but the detection of the expression level of these two genes was found to be low and inconsistent [14,39,51]. In line with this, GEF1 and GEF14 were not identified in recent transcriptomics or proteomics data [41].

The four consistently expressed GEFs have very distinct localizations, with GEF12 evenly distributed in the cytoplasm throughout the germination process, while GEF8, GEF9, and GEF11 accumulated at the site of pollen germination before pollen tube growth (Fig 1 and S1–S4 Video). A similar accumulation of GEFs was described for GEF3 and GEF14 during root hair initiation [36]. However, during root hair initiation, the polar domain requires around 30 min to be established, is locally fixed and is persistent for hours [36]. The protein accumulations we observed for GEF8, GEF9, and GEF11 were established significantly faster but shorter, as they only persisted at the pollen germination site for several minutes (Figs 1 and S2). An exciting aspect and a difference to the initiation of root hair growth is that the localization of GEF8, GEF9, and GEF11 accumulation was not fixed but could shift laterally during the initiation of pollen germination. This could be an essential feature of polar growth in pollen, as these cells need to adjust the germination site in response to contact with the papilla cells. Thus, the flexibility of this growth machinery is required to initiate growth at the optimal position in species such as *Arabidopsis thaliana* with no predetermined germination site at the pollen apertures [7]. This flexibility is not required in species with fixed pollen germination sites at the aperture, such as rice. However, also in these species, pollen-expressed ROPGEFs diversified, which indicates that multiple distinct GEF functions are required. Moreover, the Arabidopsis pollen-expressed genes are too closely related to each other, not allowing the identification of functional orthologues in other species just based on our or previous phylogeny (S1 Fig) [46].

Besides the characteristic to polarize or not, we also found differences in the timing, strength and duration of protein accumulation between GEF8, GEF9 and GEF11 (Fig 1). These differences in timing indicate that also these GEFs that show polar accumulation at the pollen germination site are not redundant in their cell biological functions and could, for example, activate different downstream proteins, as described for RIC proteins [53].

As we could already show that the pollen-expressed GEFs are not behaving redundantly in their localization and polarization, we investigated their requirement in different mutant lines. The *gef8* and *gef9* single mutants had significantly reduced pollen germination efficiencies, while *gef11* and *gef12* single mutants did not display significant reductions in pollen germination (Fig 2). Furthermore, while *gef12* did significantly enhance the *gef8;9* phenotype, which shows genetic redundancy, *gef11* did not enhance the pollen germination defects of other *gef* mutants. This shows that GEF11 has a

different function compared to GEF8 and GEF9 despite its similar localization (Figs 1 and 2). Therefore, compared to the previously postulated genetic redundancy of GEFs during pollen germination and growth [14,45], our results suggest that individual GEFs have distinct localizations and that not all GEFs act genetically redundant during pollen germination. We suggest that these GEFs activate different aspects required for polar growth. Thus, all GEFs are distinctively required for efficient pollen germination and male fertility, which only leads to a severe phenotype in higher-order mutants when multiple pathways are affected (Fig 2), similar to the observations made in loss-of-function mutants of all pollen-expressed *ROPs* [29]. Especially the opposite effect of *gef11* when combined with *gef8* or *gef9* indicates that GEF11 regulates opposing pathways that do not drive pollen tube growth but, for example, control pollen integrity, as it was shown in combination with BUPS RLKs [45]. Such non-redundant functions of GEFs within one cell type were also described in stomata, where GEF1, GEF2, and GEF4 regulate different pathways to control stomata opening or closure [23,58,59].

Additionally, to the different effects of *gef* mutants, we found stronger effects of *gef8, gef9,* and *gef12* mutants and their higher order mutants compared to previous studies. While we found *gef8;9;12* triple mutants had almost completely abolished pollen germination *in vitro* and fertility defects *in vivo*, such defects could only be observed in *gef* quadruple or quintuple mutants (Fig 2) [14,45]. The availability of suitable mutant lines can explain these discrepancies. Our observed phenotypes of *gef8* and *gef9* mutants were only evident in the CRISPR-Cas9 deletion mutants generated in this study (Figs 2 and S3). The only available T-DNA line for *GEF8* (GABI_286G01) had multiple T-DNA integrations and, in our experience, showed developmental defects independent of the integration in *GEF8*. Moreover, the available T-DNA line of *GEF9* showed a mild and less consistent phenotype than the generated CRISPR-Cas9 deletion mutant, indicating residual *GEF9* function in the T-DNA line, even if no full-length mRNA is detectable. (Figs 2 and S3).

Despite the severity of the *gef8-cΔ1;9-t1;12-t1* and *gef8-cΔ3;9-cΔ2;12-cΔ2* triple mutant phenotypes observed *in vitro*, their defect can be overcome *in vivo*, as both triple mutants, which hardly showed pollen germination on PGM, still had a significant transmission in reciprocal crosses (Figs 2C and S4), which is similar to recent data shown in *gef8;9;11;12;13* quintuple mutants [45]. Therefore, other GEFs must still be available, as total loss of ROP activity results in male infertility [29]. Potentially, other GEFs might have a higher activity *in vivo* that could rescue the lack of these primary GEFs.

The distinct localizations we observed for GEF8, GEF9, and GEF11 indicate that a specific feature of those proteins transmits this function. As the PRONE domain of all GEFs is conserved, we focused on the variable termini, which has no apparent structure (Fig 3) [32]. The predicted full-length structure of GEF8 also suggests that both termini are disordered. Thus, the confidence in the structure of these regions is very low, and it remains unclear how these termini are folded or if they can take different shapes depending on the situation. The N-terminus was shown to have activated functions for GEF activity, while the C-terminus inhibits PRONE activity [39,51]. However, in contrast to the suggested function of the terminal domains for the activity of the PRONE domain, we observed other functions for their localization. The deletion of the N-terminus did not significantly alter protein accumulation frequency or functionality but only reduced polarization strength. In contrast, the deletion of the C-terminus abolished protein accumulation and protein functionality (Fig 3C). This shows that the contribution of the termini to PRONE domain activity is different than their effect on protein localization and functionality. We hypothesize that the C-terminus of GEF8 and GEF9 is required for interaction with RLKs like PRKs, as was shown before for GEF12, as mutations in the conserved phosphorylation site can alter this accumulation (Fig 3) [14,20]. However, this interaction must be specific to GEF8 and GEF9, as we do not see such accumulations for GEF12. Additionally, the ability for protein accumulation before pollen tube emergence can be transferred onto GEF12 when exchanging the C-terminal domain (Fig 3D). Suggesting that the effect is specific to the C-terminus of GEF8 and GEF9. It is possible that mutations in this phosphorylation site of GEF8/9 affect the structure of the protein more than it does in GEF12, as this mutation in GEF12 only affects pollen tube width [39]. Furthermore, other scaffolding proteins or RLKs could be required for this specific interaction with the C-terminus of GEF8/9 and their polarization, as the previously reported PRK2, PRK6, or BUPS also interact with GEF12 [14,15,42,60,61]. Yet, it is also possible that additional mechanisms are crucial for this accumulation or that further factors transmit a specific interaction of GEF8 and GEF9 with the known RLKs at this particular time during the initiation of pollen germination. Other proteins known

to show similar accumulation at the pollen germination site before germination are the ROP effector scaffold proteins RIP, RIC, or boundary of ROP domain (BDR). However, it is unlikely that these scaffolds drive GEF accumulation, as they are ROP effectors, and their accumulation depends on ROP activation [29,62–64]. Still, these effectors may have a role in stabilizing the GEF accumulation in a feedback loop to confine the domain, as shown in root hairs [36]. Additionally, it remains unclear how crucial GEF protein polarization is, as a non-polar GEF8 with a deleted C-terminus lost functionality, while non-polar full-length GEF12 was capable of partially rescuing the *gef8* pollen phenotype (Fig 3). This indicates that the accumulation of GEFs is not essential for their function but crucial to efficiently activate specific pathways. However, it needs to be considered that GEF12 is also present at the membrane of the pollen germination site, even if it does not accumulate there. This non-polarized activity of GEF12 might be sufficient to activate ROPs if the efficient activation by polarizing GEFs is lost.

The accumulation of GEFs did not lead to any significant accumulation of ROPs at the pollen germination site (Fig 4A). Similar to GEF12, this suggests that enough ROPs are present at the plasma membrane of the pollen germination site to activate pollen germination, even if they are not accumulating at this site. This would require local activation of the non-polarized ROPs to initiate growth in a defined region. In line with this, we observed that a marker for active ROPs accumulated at the pollen germination site with a timing similar to the observed GEF accumulations. Moreover, the accumulation of the active ROP marker was significantly reduced in the *gef8-cΔ1* mutant (Fig 4C–4F). Furthermore, we found long $Ca^{2+}$ elevations before pollen germination, which differ from the described $Ca^{2+}$ oscillations in growing pollen tubes [57]. Similar to the ROP-activity marker, these $Ca^{2+}$ elevations were not lost but appeared less frequently in *gef8-cΔ1* pollen, still allowing pollen germination (Fig 4G–4I). Together, the alteration of ROP activity and $Ca^{2+}$ levels in *gef* mutants indicates that GEF accumulation is upstream of both factors and requires a different polarization mechanism. A connection to phospholipid signaling is possible, as shown in root cells [65]. Candidate proteins are phosphatidylinositol 4-phosphate 5-kinases (PIP4Ks) that regulate phospholipid abundance and regulate tip growth in pollen tubes and root hair cells [66,67]. PIP4Ks also accumulate at the pollen germination site and might act together with GEFs to drive ROP signaling, but results in root hairs suggest that PIP4K accumulation is downstream of GEF accumulation [36,67]. Studying the connection to other pathways and their regulatory connection will be a challenging task in fully understanding polarity establishment and polar growth initiation.

We show that specific GEFs are required for efficient pollen germination and male fertility. Moreover, we show that GEF8, GEF9, and GEF11 display distinct localizations compared to GEF12 and GEF13. Together, this shows that GEFs are not all redundant during pollen germination and can have specific functions within the same process in one cell. The novel polar domain of accumulated GEF8/9 protein in germinating pollen tubes was spatiotemporally flexible and not static as in previously described processes. We hypothesize that this flexibility is crucial to define the pollen germination site independently of predeterminant features and shows the *de novo* assembly of a polar growth domain.

## Materials and methods

### Plant material and growth conditions

*Arabidopsis thaliana* plants were grown on soil under long-day conditions (16 h of light at 21 °C) in a growth chamber. *Arabidopsis thaliana* ecotype Col-0 was used as a wild-type in this study.

The mutant lines *gef9-t1* (GK-717A10), *gef11-t1* (SALK_126725C), and *gef12-t1* (SALK_103614) were obtained from NASC (Nottingham Arabidopsis Stock Centre). Single t-DNA insertion was confirmed by segregation analysis, and the t-DNA insertion site was checked by sequencing the genotyping PCR product on both sides of the insertion site. Primers used for genotyping are listed in S1 Table. Double and triple mutants were made by crossing these lines and were selected by PCR. Fluorescently labelled *GEF12p*::mCit-GEF12 was used from [36].

### CRISPR/Cas9 deletion lines

To generate *gef8-cΔ1* (Δ2.1 kb)*, gef11-cΔ1* (Δ2.2 kb), and *gef12-cΔ1* (Δ1.5 kb) CRISPR/Cas9 deletion lines, we used an egg cell-specific promoter-driven CRISPR/Cas9 [68]. We used two gRNAs, one in 5′ and the other in 3′ of the gene,

cloned in tandem into one vector (S3 Fig). The selection of positive transformants in T1 generation was done on ½ MS medium containing Hygromycin (20 µg/ml).

To generate CRISPR/Cas9 deletion lines for *gef8-c*Δ*2* (Δ2.2 kb) and *gef9-c*Δ*1* (Δ2.6 kb) single mutants, *gef8c-*Δ*3/gef9-c*Δ*2* (Δ2.0 kb/Δ 2.6 kb) double mutants, as well as *gef11-c*Δ*2* (Δ2.2 kb) in *gef8c-*Δ*1/gef9-t1/*, and *gef12-c*Δ*2* (Δ1.5 kb) in *gef8c-*Δ*3/gef9-c*Δ*2* triple mutants, we used a multiplex editing approach based on an optimized zCas9i [69]. We used four gRNAs per gene, two in 5′ and two in 3′ of the gene. (S3 Fig). The selection of positive lines in T1 was performed using seed fluorescence and Basta resistance. T2 selection was made by selecting non-glowing seeds. gRNAs were determined using ChopChop (https://chopchop.cbu.uib.no/) [70] and CCTop (https://cctop.cos.uni-heidelberg.de/) [71] and successful cloning was confirmed by sequencing. Genotyping was performed by primers 200–300 bp outside the deleted area. The deletion was characterized by sequencing the resulting PCR product. Confirmation of homozygous mutant alleles was done by combination with a primer binding in the deleted region and was confirmed in the following generation. All primers are listed in S1 Table, and details of all mutant sequences are shown in S3 Fig.

### Plasmid construct

All non-CRISPR/Cas9 constructs were generated using the GreenGate cloning system [72] with modified cloning procedures. A list of combined modules and their sources is provided in S1 Table. As native promoter sequences upstream regions of the START codon, including the 5′UTRs, were cloned for *GEF8* (AT3G24620, −2,501 bp), *GEF9* (AT4G13240, −1,463 bp), *GEF11* (AT1G52240, −973 bp), *GEF12* (AT1G79860, −701 bp), and *GEF13* (AT3G16130, 850 and 1,200 bp) The *LAT52* promoter was amplified from LHR lines [4] and a RGeco1 module was kindly provided by Rainer Waadt [73] The CRIB4 module was generated by amplification of the CRIB domain of RIC4 (AT5G16490, amino acid I64-I131) as described before [54] from flower cDNA. All primers used to generate new Entry-Vector modules are listed in S1 Table. The correct amplification and cloning of entry-vector modules were confirmed by sequencing. We generated a new GreenGate-compatible destination vector to achieve a higher plant transformation efficiency. For this, we amplified the vector backbone of pHEE401E [68] and the GreenGate cloning cassette of pGGZ003 [72] by PCR (Primers in S1 Table) and combined both fragments using added MluI and BamHI sites. The resulting plasmid was confirmed by sequencing and named pGGX000.

### Phenotyping of pollen germination efficiency

For phenotyping of *in vitro* pollen germination efficiency, pollen of freshly opened flowers was germinated at 22 °C on solid pollen germination medium (PGM), containing 1.5% agarose (w/v), 18% sucrose (w/v), 0.01% boric acid (w/v), 1 mM $CaCl_2$, 1 mM $Ca(NO3)_2$, 1 mM $MgSO_4$, 10 µM 24-epibrassinolides (MedChemExpress, HY-N084824, dissolved to 5 mM in Ethanol), and adjusted to pH = 7.0 using 100 mM KOH) [47]. Germination was analyzed 4 h after imbibition on the PGM, using a Leica MZ16 stereomicroscope equipped with DMC5400. The images were analyzed using the multipoint tool of ImageJ (https://imagej.nih.gov/ij/docs/guide/146-19.html#sec:Multi-point-Tool), then making the ratio of germinated and ungerminated pollen to get the germination efficiency. Each line was analyzed in at least three independent experiments, including three replicates in each experiment. Each data point represents an independent replicate analyzing more than 150 pollen grains. To test for significant differences in pollen germination efficiency, a one-way ANOVA ($p$-value < 0.0001) with a post-Tukey test (significance level < 0.05) was performed in GraphPad Prism 9.

### Fluorescence imaging

Imaging of *in vitro* pollen germination was done at 22 °C on solid PGM, containing 1.5% agarose (w/v, 10% sucrose (w/v), 0.01% boric acid (w/v), 5 mM $CaCl_2$, 5 mM KCl, 1 mM $MgSO_4$, adjusted to pH = 7.5 using 100 mM KOH [48] and supplemented with 10 µM 24-epibrassinolides (MedChemExpress, HY-N084824, dissolved to 5 mM in Ethanol) [47].

Live-cell imaging was performed using an Olympus confocal FV-1000 equipped with a 40× water immersion (1.15 NA) objective, an Argon laser, and a 559 nm diode laser. Signals were detected with high-sensitivity detectors. mCitrine (YFP) was excited at 515 nm and detected between 520–550 nm. mScarlet (RFP) and RGeco signals were exited at 559 nm and were detected between 580 and 653 nm. The pinhole was set to 1AU, and images were taken with a 2× line average in a 1024 × 1024 pixel scanning field. The same settings were applied to all images, and the excitation laser intensity was set to a minimum for the individual lines to avoid phototoxicity. GEF pollen germination images were acquired every 30 s for 30–60 min. For Lat52p::RGeco, images were acquired every 5 s for 30 min.

### Image analysis and quantifications

All images were processed and analyzed using ImageJ (FIJI), and values of measurements were processed in Excel. Box plots were generated using BoxPlotR (shiny.chemgrid.org/boxplotr/). Images were registered with the MultiStackRegistration plugin to correct for drift in time-laps experiments.

Kymographs were generated with the Multiple Kymograph plugin, using a line width of nine pixels for kymographs across the pollen grain and five pixels for kymographs around the pollen grain. The "Royal" Lookup-Table to false-color Kymographs and highlight signal intensity differences.

Quantification of polarization was done by calculating the ratio of the signal intensity at the pollen germination site (15 × 5 µm) and a region in the pollen grain center (15 × 5 µm) for each time point. To adjust for expression differences between individual pollen, these ratios were normalized to the average of five consecutive time points without polarization before pollen germination. To determine protein polarization above background, GEF12 was used as a reference, and a protein was considered polar if the intensity ratio was higher than twice the standard deviation over the average GEF12 value for more than a single time point. Polarization frequency (portion of pollen with polar protein) and maximum polarization values were extracted using Excel. To focus on the polarization of proteins at germination initiation, values within 1.5 min before germination were not considered for this analysis. To test for significant differences in maximum polarization values, a one-way ANOVA ($p$-value < 0.0001) with a post-Tukey test (significance level < 0.05) was performed in GraphPad Prism 9. To test for significantly different polarization frequencies, $X^2$ tests with GEF8 or GEF9 as reference were performed (significance level < 0.05) in Excel.

$Ca^{2+}$ levels were analyzed in a 5 × 5 µm region at the pollen germination site and normalized to a 15 × 5 µm region at the opposite side of the pollen grain. The resulting ratios were corrected for slow and constant signal changes (e.g., bleaching) by normalizing to a sliding 20-frame average. $Ca^{2+}$ increases were considered elevations above the background if the normalized values elevated over a threshold of 5× standard error of means over the average of the individual measurement. The number of elevations was identified manually in Excel and shown in S7 Fig.

### Supporting information

**S1 Fig. Phylogeny of all ROPGEFs of *Arabidopsis thaliana* and expression levels in mature pollen. (A)** Phylogenetic tree of the 14 ROPGEFs of *Arabidopsis thaliana* after alignment of the full-length protein sequence in Jalview, using the integrated MUSCLE alignment tool and calculation of an average distance (BLOSUM62) tree. **(B)** Presence of transcript or protein for all 14 *ROPGEFs* in mature pollen, according to the ATHENA – Arabidopsis THaliana Expression Atlas [41]. GEFs are sorted according to the phylogenetic tree. Levels of the transcript are shown in transcripts per kilobase million (TPM) and intensity-based absolute quantification (iBAQ) for protein levels. NA indicates that no transcript or protein was detected in this tissue.
(PDF)

**S2 Fig. mCit-GEF13 is not detectable in pollen and does not accumulate at the pollen germination site. (A)** Example images of mCit-GEF13 under the control of a *GEF13* promoter fragment in mature pollen grains and pollen tubes

grown through a cut pistil. In neither case could any signal be detected. **(B)** Representative localization of mCit-GEF13 under control of a *GEF12* promoter fragment during pollen germination. Timepoint 0 corresponds to the beginning of pollen tube emergence, and arrowheads mark the site of pollen emergence. All scale bars represent 10 μm.
(PDF)

**S3 Fig. Genomic structure and mutant allele information of *GEF8, GEF9, GEF11* and *GEF12*. (A–D)** *GEF8, GEF9, GEF11*, and *GEF12* genomic structures with the corresponding gRNA sites (scissors) and T-DNA insertion sites (arrowhead) for *gef9-t1* (GK-717A10), *gef11-t1* (SALK_126725C), and *gef12-t1* (SALK_103614). Promoter regions are shown as white boxes, UTRs in cyan, and exons in grey boxes. WT sequence for each gene and corresponding CRISPR/Cas9 deletion line is shown, and the size of CRISPR/Cas9 induced deletions is indicated. The gRNA target sequence is highlighted in color with the PAM in bold. The START and STOP codons are underlined. **(E–G)** Genomic structures with T-DNA insertion sites (arrowhead) for *gef9-t1* (GK-717A10, **E**), *gef11-t1* (SALK_126725C, **F**), and *gef12-t1* (SALK_103614, **G**) with the location of primers used to test the mRNA presence. The tables show the expected PCR product size with the indicated primer combination for non-spliced templates (genomic) and correctly spliced templates (CDS). The gel images show PCR products of PCRs using the indicated primer combination on cDNA from Arabidopsis flowers of Col-0 in comparison to *gef9-t1* **(E)**, *gef11-t1* **(F)**, or *gef12-t1* **(G)**.
(PDF)

**S4 Fig. Reciprocal crosses of *GEF* mutant lines.** Quantification of mutant allele frequency in F1 generation of reciprocal crosses with Col-0 as female and the indicated mutants as pollen donors (left) or indicated mutants as female and Col-0 as pollen donor (right). The heterozygous allele of each genotype is shown in bold. Asterisks indicate a significant difference in the allele frequency from the expected 50% according to $X^2$-test ($p < 0.05$). For underlying data of all quantification see S1 Data.
(PDF)

**S5 Fig. Prediction of full-length GEF8-ROP1 complex. (A)** Full-length GEF8 and ROP1 protein structures predicted separately by AlphaFold2 (ColabFold v1.5.5) as shown in Fig 3A, and the most likely structure model is shown in different angles [74,75]. The color coding of GEF8 represents the Predicted Local Distance Difference Test (pLDDT) value of the model structure confidence, with blue representing high confidence and orange low confidence. While the conserved PRONE domain has high confidence, the N- and C-terminal regions have very low confidence and contain likely disordered regions. Thus, their structure needs to be taken with caution. **(B)** pLDDT value per position of all five models. All models have a similar likelihood in the PRONE domain and the terminal regions. **(C)** The individual predicted structures of GEF8 are color-coded using the pLDDT value, and the terminal amino acids are colored magenta (N) or cyan **(C)**. Below each model, the Predicted aligned error (PAE) data should be considered to assess the domain accuracy of the predicted structures. On the right, the five models are superimposed on each other to highlight the differences. While the PRONE domain largely overlays in all models, the position of the terminal regions and their orientation is different in all models.
(PDF)

**S6 Fig. The C-terminus of GEF8 and GEF9 and its conserved phosphorylation site are required for protein accumulation at the pollen germination site. (A–H)** Individual relative fluorescence intensity profiles at the pollen germination site of different GEF mutant constructs of the measurements shown and quantified in Fig 3. Thin yellow lines show individual measurements, and thick yellow lines represent the average of all samples. As a reference, the average intensity profiles of GEF8 or GEF9 are shown in black, and that of GEF12 as a dotted line. **(A)** GEF8p::mCit-GEF8$^{\Delta N}$ (*n* = 10), **(B)** GEF8p::mCit-GEF8$^{\Delta C}$ (*n* = 13), **(C)** GEF9p::mCit-GEF9$^{\Delta N}$ (*n* = 16), **(D)** GEF9p::mCit-GEF9$^{\Delta C}$ (*n* = 14), **(E)** GEF12p::mCit-GEF12$^{GEF8C}$ (*n* = 13), **(F)** GEF12p::mCit-GEF12$^{GEF9C}$ (*n* = 13), **(G)** GEF8p::mCit-GEF8$^{S518A}$ (*n* = 21), **(H)** GEF8p::mCit-GEF8$^{S518D}$ (*n* = 13). **(I)** Representative localization of the GEF8 C-terminus (*Gef8p*::mCit-GEF8$^{CTerm}$) during

pollen germination. Of the 15 observed germinated pollen, of which 4 could be observed for 15 min before germination, none showed any polarization or protein accumulation. Timepoint 0 corresponds to the beginning of pollen tube emergence, and arrowheads mark the site of pollen emergence. Scale bars represent 10 μm. For underlying data of all quantification see S1 Data.
(PDF)

**S7 Fig. Individual measurements of mCit-CRIB4 and RGeco1 at the pollen germination site. (A)** Individual relative fluorescence intensity profiles at the pollen germination site of Gef12p::CRIB4-mCit in Col-0 (yellow, $n$ = 12) or *gef8-cΔ1* (black, $n$ = 12) background, as they are shown and quantified in Fig 4. Thin lines show individual measurements, and thick lines represent the average of all samples. **(B, C)** Normalized intensity plots of Lat52::RGeco1 in Col-0 (B, $n$ = 13) or *gef8-cΔ1* (C, $n$ = 7) background. Black lines represent the normalized RGeco signal intensity. Magenta lines show the significance threshold used to define large $Ca^{2+}$ elevations. The top graphs correspond to the measurement shown in Fig 4. For underlying data of all quantification see S1 Data.
(PDF)

**S1 Video.** Gef8p::mCit-GEF8, corresponding to Fig 1A.
(AVI)

**S2 Video.** Gef9p::mCit-GEF9, corresponding to Fig 1B.
(AVI)

**S3 Video.** Gef11p::mCit-GEF11, corresponding to Fig 1C.
(AVI)

**S4 Video.** Gef12p::mCit-GEF12, corresponding to Fig 1D.
(AVI)

**S5 Video.** Gef12p::mCit-GEF13, corresponding to S2B Fig.
(AVI)

**S6 Video.** Gef8p::mCit-GEF8$^{ΔN}$, corresponding to Fig 3C.
(AVI)

**S7 Video.** Gef8p::mCit-GEF8$^{ΔC}$, corresponding to Fig 3C.
(AVI)

**S8 Video.** Gef9p::mCit-GEF9$^{ΔN}$, corresponding to Fig 3C.
(AVI)

**S9 Video.** Gef9p::mCit-GEF9$^{ΔC}$, corresponding to Fig 3C.
(AVI)

**S10 Video.** Gef12p::mCit-GEF12$^{GEF8C}$, corresponding to Fig 3D.
(AVI)

**S11 Video.** Gef12p::mCit-GEF12$^{GEF9C}$, corresponding to Fig 3D.
(AVI)

**S12 Video.** Gef8p::mCit-GEF8$^{CTerm}$, corresponding to S6I Fig.
(AVI)

**S13 Video.** Gef8p::mCit-GEF8$^{S518A}$, corresponding to Fig 3E.
(AVI)

**S14 Video.** Gef8p::mCit-GEF8$^{S518D}$, corresponding to Fig 3E.
(AVI)

**S15 Video.** Rop1p::mCit-ROP1, corresponding to Fig 4A.
(AVI)

**S16 Video.** Rop3p::mCit-ROP3, corresponding to Fig 4A.
(AVI)

**S17 Video.** Rop5p::mCit-ROP5, corresponding to Fig 4A.
(AVI)

**S18 Video.** Gef12p::CRIB4-mCit in Col-0, corresponding to Fig 4C.
(AVI)

**S19 Video.** Gef12p::CRIB4-mCit in *gef8-cΔ1*, corresponding to Fig 4C.
(AVI)

**S20 Video.** Lat52p::RGeco1 in Col-0, corresponding to Fig 4G.
(AVI)

**S21 Video.** Lat52p::RGeco1 in *gef8-cΔ1*, corresponding to Fig 4G.
(AVI)

**S1 Table.** Primer and plasmid information.
(XLSX)

**S1 Data.** Quantitative data and statistics.
(XLSX)

## Acknowledgments

We are grateful to Andrea Bleckmann, Claus Schwechheimer (Technical University of Munich), and Guido Grossmann (University Düsseldorf) for helpful comments and suggestions on our manuscript. We thank Thomas Dresselhaus (University Regensburg) for the possibility of confirming our results on an independent microscope setup and Andrea Bleckmann for imaging advice and microscope instructions. We acknowledge the support of the Center for Advanced Light Microscopy (CALM) of the TUM School of Life Sciences.

## Author contributions

**Conceptualization:** Philipp Denninger.

**Data curation:** Philipp Denninger.

**Formal analysis:** Alida Melissa Bouatta, Lisa Riederauer.

**Funding acquisition:** Philipp Denninger.

**Investigation:** Alida Melissa Bouatta, Franziska Anzenberger, Lisa Riederauer, Philipp Denninger.

**Methodology:** Alida Melissa Bouatta, Philipp Denninger.

**Project administration:** Philipp Denninger.

**Resources:** Alida Melissa Bouatta, Franziska Anzenberger, Andrea Lepper, Philipp Denninger.

**Supervision:** Philipp Denninger.

**Visualization:** Alida Melissa Bouatta, Philipp Denninger.

**Writing – original draft:** Alida Melissa Bouatta, Philipp Denninger.

**Writing – review & editing:** Alida Melissa Bouatta, Andrea Lepper, Philipp Denninger.

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
