## [Editor Report · Decision Letter 0]

21 Jul 2024

Dear Dr Denninger, 

Thank you for submitting via Review Commons your manuscript entitled "Polarised subcellular activation of ROPs by specific ROPGEFs drives pollen germination in Arabidopsis thaliana" for consideration as a Research Article by PLOS Biology.

Your manuscript has now been evaluated by the PLOS Biology editorial staff as well as by an academic editor with relevant expertise and I am writing to let you know that we would like to invite you to submit a revision of your manuscript.

However, before we can do that, we need you to complete your submission by providing the metadata that is required for full assessment. To this end, please login to Editorial Manager where you will find the paper in the 'Submissions Needing Revisions' folder on your homepage. Please click 'Revise Submission' from the Action Links and complete all additional questions in the submission questionnaire.

Once your full submission is complete, your paper will undergo a series of checks in preparation for peer review. After your manuscript has passed the checks it will send you the decision. To provide the metadata for your submission, please Login to Editorial Manager (https://www.editorialmanager.com/pbiology) within two working days, i.e. by Jul 23 2024 11:59PM.

Kind regards,

Ines

--

Ines Alvarez-Garcia, PhD

Senior Editor

PLOS Biology

---

## [Editor Report · Decision Letter 1]

25 Jul 2024

Dear Dr Denninger,

Thanks again for submitting via Review Commons your manuscript entitled "Polarised subcellular activation of ROPs by specific ROPGEFs drives pollen germination in Arabidopsis thaliana" was peer-reviewed at PLOS Biology.

It has now been evaluated by the PLOS Biology editors and an Academic Editor expert on the field, and we would like to invite you to revise the work to thoroughly address the reviewers' reports as you have indicated in your revision plan. Given the extent of revision needed, we cannot make a decision about publication until we have seen the revised manuscript and your response to the reviewers' comments. Your revised manuscript is likely to be sent for further evaluation by all or a subset of the reviewers.

We expect to receive your revised manuscript within 3 months. Please email us (plosbiology@plos.org) if you have any questions or concerns, or would like to request an extension. At this stage, your manuscript remains formally under active consideration at our journal; please notify us by email if you do not intend to submit a revision so that we may withdraw it.

**IMPORTANT - SUBMITTING YOUR REVISION**

3. Resubmission Checklist

a) *PLOS Data Policy*

b) *Published Peer Review*

Thank you again for your submission to our journal and please don't hesitate to contact us if you have any questions or comments.

Sincerely,

Ines

--

Ines Alvarez-Garcia, PhD

Senior Editor

PLOS Biology

---

## [Decision Letter · Decision Letter 2]

21 Feb 2025

Dear Dr Denninger,

Thank you for your patience while we considered your revised manuscript entitled "Polarised subcellular activation of ROPs by specific ROPGEFs drives pollen germination in Arabidopsis thaliana" for publication as a Research Article at PLOS Biology. This revised version of your manuscript has been evaluated by the PLOS Biology editors, the Academic Editor and the three original reviewers from Review Commons.

Based on the reviews, we are likely to accept this manuscript for publication, provided you satisfactorily address the data and other policy-related requests stated below.

In addition, we would like you to consider a suggestion to improve the title:

"Polarized subcellular activation of Rho proteins by specific GEFs drives pollen germination in Arabidopsis thaliana"

We expect to receive your revised manuscript within two weeks. 

*Published Peer Review History*

*Press*

Sincerely,

Ines

--

Ines Alvarez-Garcia, PhD

Senior Editor

PLOS Biology

Fig. 1C-I; Fig. 2A, C; Fig. 3F-H; Fig. 4E, F, I; Fig. S4; Fig. S5B; Fig. S6A-H and Fig. S7A-C

CODE POLICY

Per journal policy, if you have generated any custom code during the course of this investigation, please make it available without restrictions. Please ensure that the code is sufficiently well documented and reusable, and that your Data Statement in the Editorial Manager submission system accurately describes where your code can be found. [IF APPLICABLE: As the code that you have generated to XXX is important to support the conclusions of your manuscript, its deposition is required for acceptance.]

Reviewers' comments

Rev. 1:

The authors have fully addressed my questions and the manuscript is substantially improved. This study greatly advanced our understanding of GEFs in cell polarization. I recommend it for publication.

Rev. 2:

The authors have addressed most of my comments.

Rev. 3:

The revised manuscript is significantly improved and I feel that the authors have thoroughly addressed my (as well as other reviewers') concerns and suggestions in a satisfactory manner. While I initially suggested in vivo imaging of pollen germination on stigmatic papillae, I agree this is beyond the scope of the current manuscript. The other additional experiments compensate for this limitation. Overall, I believe the revised manuscript represents a significant advancement in understanding ROPGEF function in pollen germination and polarity establishment. Given the significant refinements and robustness of the presented data, I have no further requests for experiments or textual changes. The work is of high quality and will be of great interest to the plant science community. I congratulate the authors on the excellent work and look forward to seeing it published.

---

## [Editor Report · Decision Letter 3]

2 Apr 2025

Dear Dr Denninger,

Thank you for the submission of your revised Research Article entitled "Polarised subcellular activation of Rho proteins by specific ROPGEFs drives pollen germination in Arabidopsis thaliana" for publication in PLOS Biology. On behalf of my colleagues and the Academic Editor, June Nasrallah, I am delighted to let you know that we can in principle accept your manuscript for publication, provided you address any remaining formatting and reporting issues. These will be detailed in an email you should receive within 2-3 business days from our colleagues in the journal operations team; no action is required from you until then. Please note that we will not be able to formally accept your manuscript and schedule it for publication until you have completed any requested changes.

PRESS

Sincerely, 

Ines

--

Ines Alvarez-Garcia, PhD

Senior Editor

PLOS Biology
